# Flow Analysis Based on Cathodic Current Using Different Designs of Channel Distribution In PEM Fuel Cells

**Marco Antonio Zamora-Antuñano** [1,*,†], **Pablo  Esaú Orozco Pimentel** [2],
**Germán Orozco-Gamboa** [2], **Raul García-García** [3], **Juan Manuel  Olivárez-Ramírez** [3],
**Edrei Reyes Santos** [1] **and  Álvaro De Jesús Ruiz Baltazar** [4,†]

1   Engineering Area of the Universidad del Valle de México, Campus Querétaro, Naranjos Punta
    Juriquilla 1000, Santa Rosa Jáuregui, Santiago de Querétaro, Qro. 76230, Mexico
2   Centro de Investigación y Desarrollo Tecnológico en Electroquímica (CIDETEQ), Parque Tecnológico
    Querétaro-San Fandila, Pedro Escobedo, Qro. 76703, Mexico
3   Academic Division of Chemistry and Renewable Energies of the Universidad Tecnológica de San Juan del
    Río. Av. La Palma No 125 Vista Hermosa, 76800 San Juan del Río, Qro. 76800, Mexico
4   Nanotechnology Engineering Area: Cátedra CONACyT, CFATA-UNAM, Campus UNAM-Juriquilla,
    Boulevard Juriquilla No. 3001, Santiago de Querétaro, Qro. 76230, Mexico
*   Correspondence: marco.zamora@uvmnet.edu; Tel.: +52-4422111900 (ext. 11218)
†   These authors contributed equally to this work.

**Abstract:** In this work, a physical and numerical simulation of cathodic current for different designs of the channel distribution in PEM fuel cells was carried out. The first design consisted serpentine-type channels with abrupt changes in flow direction. On the other hand, Designs 2 and 3 were made of serpentine channels with a more gradual change in flow direction. The fourth design was a crisscross-type channel, which was based on continually redirecting the flow, while Design 5 was made with straight parallel channels. Designs 1–3 had one intake, while Designs 4 and 5 had three. The latter two produced more uniform electrical current distributions than Designs 1–3. It can be concluded that the intakes situated effectively within each design were as important as the shape of the channel configuration. Finally, the parallel channel flow field (Design 5) was the best alternative for current collectors due to its better performance.

**Keywords:** electrical current distributions; fuel cell; design of distribution channels

## 1. Introduction

There is no consensus about the projected date of global peak-oil production due to the lack of accurate technology to determine the stock of current world oil supplies. However, if the rate of production and consumption of the Mexican oil supplies remains unchanged, then the Mexican reserves might be depleted before 2030 [1]. Consequently, in the Mexican energy market, it is necessary to incorporate renewable energy supplies combined with fuel cell technology. The process of developing fuel cell technology has not been applied optimally in Mexico, and despite scientific advances, its development is still very expensive, so universities and research centers mainly use simulation as an alternative that allows them to experiment and develop proposals for scientific and technological applications in the field of fuel cells. In this work, simulation was used to compare the behavior of a fuel cell with different distribution channels. Actually, a large number of companies, education institutions, and research centers are currently developing research programs for fuel cells. However, there is still a significant amount of technical challenges that must be addressed,

such as the choice and handling of fuels, the lack of infrastructure for hydrogen storage, the high cost of this, etc. The technology of fuel cells is sufficiently developed for its commercialization, except for the cost, which is still very high. Therefore, the most important technological and scientific activity in the development of fuel cells is aimed at reducing costs and improving their performance. The development and application of new configurations of collector plate geometries is a fundamental aspect to improve the efficiency of fuel cells. In the literature, many researchers continually report new findings and proposals for improvement or new geometries, which allow different aspects of the technology and fuel cell efficiency to be improved, but they are still not enough to reduce the costs involved in the development of this technology. In the following years, it is expected to reduce the cost of manufacturing fuel cells so that they are economically viable and mass production can occur, at least. In the short term, it is expected that fuel cell application will be present in the power generation and remote power distribution systems.

The essential components of a proton exchange membrane fuel cell (PEMFC) are an anodic catalyst layer, an electrolyte, a cathodic catalyst layer, and current collectors. Additionally, gaskets for preventing gas leakage between the anode and the cathode have to be considered. Current collectors, which also provide the conduits for gas distribution, are bipolar plates in a stack; they occupy over 90% of the volume and contribute 80% of the mass of a fuel cell stack. Flow fields can be effectively designed through the optimization of their configuration and shape, thus resulting in improved current collectors. This is desirable for achieving higher electrical power. Therefore, the study of various plate designs was performed by simulating the processes within a PEMFC.

Three different versions of serpentine gas flow field design were investigated because it is usually the reference of choice in the literature. Some of the main points about the performance of fuel cells can be mentioned: (a) for fuel cell energy systems, the development of appropriate thermal management is a fundamental issue [2]; (b) the flowfield channel of a PEM fuel cell can enhance the mass transfer of reactant gas from the channel into the catalyst layer and improve cell performance [3]; (c) the design of the geometries is fundamental in the operation of the fuel cell [4]; (d) fuel cell technology is an alternative in the use of alternative energy [5]; (e) companies such as Toyota Motor Corporation (TMC) have been developing fuel cell (FC) technology since 1992; TMC created a new fuel cell in 2008 that improved the removal of water and gas diffusion by adopting a newly-developed three-dimensional (3D) fine-mesh flow field at the cathode, and through the use of fuel cells, the performance in car engines has been improved [6,7]; (f) the pressure drop is one of the factors that influences the overall performance of the cell, both directly and indirectly through the interaction with other factors, including water management in the cell; different designs were analyzed through computational fluid dynamics (CFD), and the effect of varying the flow rate on the pressure drop for each of the designs modeled was studied [8]; (g) the flow distribution obtained with three different bipolar plate geometries has been studied, analyzing their fluid dynamic performance. Three plate topologies were selected in a way representative of different design models commonly used from experimental and numerical simulations, and using simulation, it was possible to determine the most optimal channel geometry [9]. Our versions consisted of typical serpentine flow channel design, which has a cross-section of around 1 mm$^2$ and a length of up to a few meters, with many changes in flow direction. The fourth design consisted of a crisscross pattern [8,9]. This latter design was first proposed by Barreras [4] and modified in our previous work [10]. This design was selected for study because it reduces the blockage problem caused by the accumulation of water droplets on the cathode side, which occurs in serpentine designs. The last design consisted of parallel straight-line channels in which the flow direction remained unchanged. The main focus of this study is to measure the electrical current distribution in five different designs of bipolar plate flow channels to find an efficient and a commercially-viable PEMFC. We determined the best option for the possible development of a prototype of a fuel cell at the laboratory level from the results achieved. The serpentine flow channel is one of the most common channel designs for PEMFC [10], and it was selected as the reference for comparison of the new

geometries. Table 1 shows the characteristics of the geometries. In the present work, the term "design" refers to the shape of the geometries of the bipolar plates of the PEM fuel cell.

**Table 1.** Characteristics of the flow channel designs in the bipolar plates.

| Design No. | Geometry Type | Characteristics of Each Channel Design of Bipolar Plate Geometry |
|---|---|---|
| 1 | Serpentine | Consists of serpentine channels with abrupt changes in flow direction. |
| 2 | Serpentine | Consists of serpentine channels; the direction of the flow was changed gradually. |
| 3 | Serpentine | Consists of serpentine channels with modified dimensions regarding Designs 1 and 2 |
| 4 | Crisscross | This design is based on continually re-directing the flow in a crisscross pattern. |
| 5 | Parallel | This design has parallel and straight channels, and each channel must be fed with air at a constant pressure. |

## 2. Methodology

Fluid flow and mass transfer in the gas-distribution plates were modeled numerically from a macro-scale perspective. Computational fluid dynamics (CFD) software provided by Multiphysics COMSOL® v.3.4 was used to simulate the resulting electrical current distributions in the five flow field designs. The simulation began when a flow field channel design was drawn using the computer-aided design (CAD) function of this software. The geometric model consisted of three regions: the gas-flow channels, a gas-diffusion layer, and a cathode. In the gas diffusion layer, such as a porous carbon cloth, only diffusional processes occurred, and in the cathode, the electrochemical reaction took place.

In the following steps according to Figure 1, the assumptions made in the fuel cell modeling were: (1) a three-dimensional (3D) and a single-phase mode; (2) air and water vapor were considered ideal gases; (3) the flow was laminar, steady, and incompressible; (4) the electrical power behavior of the cell was kinetically controlled by an oxygen reduction reaction (ORR); (5) the cathode and gas-diffusion layer were isotropic and homogeneous; (6) the ohmic potential drop in the current collector and gravity were negligible; (7) the process was isothermal; and (8) the gas phase diffuses in the $x$, $y$, and $z$ axes.

The equations employed for our mathematical simulations of the fuel cell are listed in Table 1, and their scalar equations are shown in Table 2. The scalar equations are listed in Table 3. Table 4 details boundary conditions , and the data came from the literature [11,12], including the molar diffusion volumes [13], the exchange current density [14], and the gas diffusion layer thickness Equation (2). The Navier–Stokes Equation (1) was solved first in order to determine the gas velocity field in the channels. This equation was subjected to boundary conditions, Equations (5)–(9). Besides, the gas species (i.e., $N_2$, $O_2$, $H_2O$) were taken into consideration. Therefore, a multi-component diffusion (Stefan–Maxwell) Equation (2) was needed to solve the diffusion layer. The Stefan–Maxwell Equation (2) needs to incorporate scalar relations for fluid properties [15,16]. Furthermore, Equation (2) was solved considering the convective flux Equation (10). Meanwhile, Equation (10) was solved using the velocity in the pore [17] and mass flux vectors [18,19], and these latter equations were determined by electrode-reaction kinetics [20].

The CFD software employs a finite-element method for solving the partial differential equations (PDE). This method involves dividing the domain of the PDEs of interest into a finite number of linear elements (i.e., the "mesh"). The triangular mesh offers better conformity to curvature than the square elements, which is used mostly in areas with right angles, since it reduces the number of mesh elements required. The program allows refining of the mesh, carefully adjusting the type and size of elements on any curvature of the geometry. A mesh consisting of 18,000 computational elements was applied to the domain (see Figure 1D). The iterative computations were terminated once the value of the residues fell to less than $1 \times 10^{-6}$.

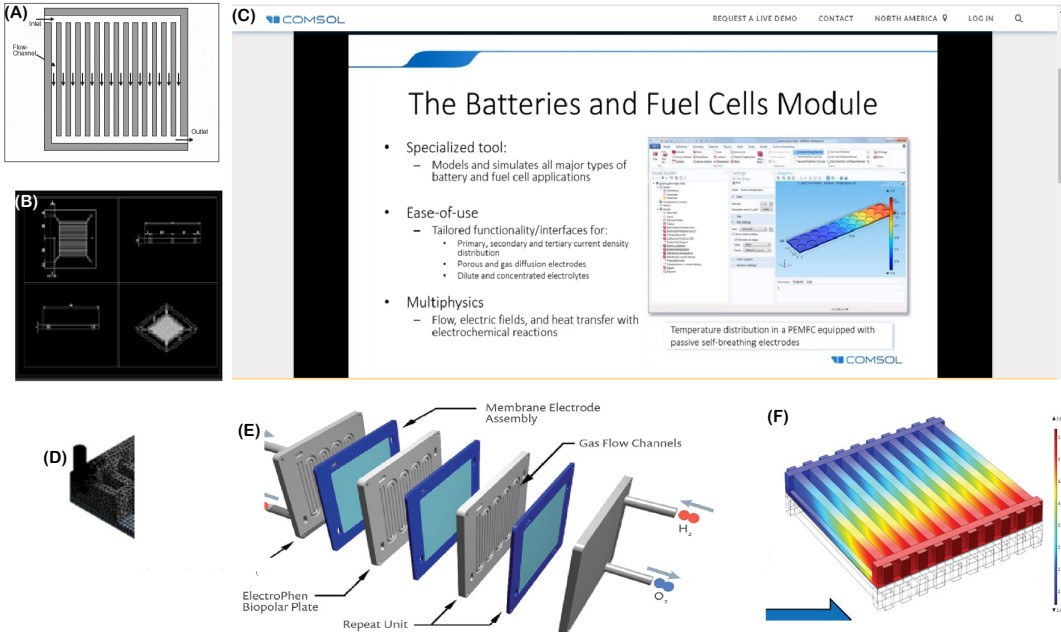

**Figure 1.** (**A**) Concept of the geometries of bipolar plates. (**B**) Design of the geometries in AutoCAD (CAD). (**C**) Three-dimensional (3D) view in Multiphysics COMSOL®v.3.4. (**D**) Figure mesh. (**E**) Isometry and components of the PEM fuel cell. (**F**) Simulation of variables according to the parameters. In (A,E,F), we can see the shape of the channels.

A computer with a dual-core processor at a speed of 3.0 GHz and 8 GB of RAM was used. The parametric nonlinear numerical method and further details were described extensively by Irshad [8] and Chen [21].

**Table 2.** Governing equations.

| Name | Equations | Number |
|---|:---:|:---:|
| Navier–Stokes Laminar Flow Stefan–Maxwell | $\rho \dfrac{\partial u}{\partial t} - \nabla \bullet \mu \left( \nabla u + (\nabla u)^T \right) + \rho u \bullet \nabla u + \nabla p = 0$ | (1) |
| Diffusion and Convection | $\nabla \left[ -\rho \omega_i \sum_{j=1}^{N} D_{ij}^{eff} \dfrac{M}{M_j} \left( \nabla \omega_j + \omega_j \dfrac{\nabla M}{M} \right) + \rho \omega_i u \right]$ | (2) |

**Table 3.** Scalar equations.

| Name | Equations | Number |
|---|:---:|:---:|
| Stefan–Maxwell Diffusion Coefficient | $D_{ij} = k \dfrac{T^{1.75}}{p \left( v_i^{1/3} + v_j^{1/3} \right)^2} \left[ \dfrac{1}{M_i} + \dfrac{1}{M_j} \right]$ | (3) |
| Stefan–Maxwell Effective Diffusion | $D_{ij}^{eff} = D_{ij} \varepsilon^{1.5}$ | (4) |
| Total Molar Mass | $M = \sum_{i=1}^{n} M_i \omega_i$ | (5) |
| Electrical Current Density | $i_c = -S_a \delta i_0 \dfrac{\omega_j}{\omega_{j,0}} e^{\left( \frac{\eta F}{2RT} \right)}$ | (6) |

The equations are written in indicial-tensor notation with: $\rho$, density kg/m$^3$; t, time s; $\mu$ dynamic viscosity of oxygen $= 2 \times 10^{-5}$ Pa-s (5); $u$, velocity vector m/s; p, pressure Pa; $\omega_i$ weight fraction of the i$^{th}$ species; with an inlet mass fraction $O_2 = 0.1447$, a mass fraction of $H_2O = 0.3789$, and a mass fraction of $N_2 = 0.4764$; $D_{eff,ij}$, the effective binary diffusion coefficient (see Equation (4) in [22–24]); $M$, the total molar mass of the mixture g/mol (see Equation (5) in [19–21,25–27]); $M_j$, the molecular weight of species $j$, g/mol; $MO_2$, the molecular weight of $O_2 = 32$ g/mol; $M_{H2O}$, the molecular weight of $H_2O = 18$ g/mol; $M_{N2}$, the molecular weight of $N_2 = 28$ g/mol; $D_{ij}$, the binary diffusion coefficient for species $i$ and $j$; $k$, the Maxwell diffusion constant $= 3.16 \times 10^{-8}$ Pa·m$^2$/s; T, temperature $= 353$ °K; $v_i$, the molar diffusion volume of the j$^{th}$ species, cm$^3$/mol; $v_{O2}$, the molar diffusion volume of $O_2 = 16.6 \times 10^{-6}$ cm$^3$/mol; $vH_2O$, molar diffusion volume of $H_2O = 12.7 \times 10^{-6}$ cm$^3$/mol; $vN_2$, the molar diffusion volume of $N_2 = 17.9 \times 10^{-6}$ cm$^3$/mol; for $M_j$, $MO_2$, $MH_2O$, and $MN_2$, see Table 1; $\varepsilon$, porosity $= 0.5$; $i_c$, cathode current; $S_a$ specific surface area $= 1 \times 10^7$m$^2$/m$^3$; $\delta$, active layer thickness $= 10$ um; $i_0$, exchange current density $= 1.06 \times 10^{-6}$ mA/cm$^2$; F, Faraday's constant $= 96485$ C/mol, R, gas constant $= 8.314$ J/(mol·K); $\eta$ overpotential between 0.2 and 0.82 V; $t_{H_2O}$ electro-osmotic drag $= 3$; gdl, gas diffusion layer thickness $= 0.2$ mm.

Five different air flow field geometries adjacent to a PEMFC oxygen cathode were compared by numerical simulations. Table 5 details the dimensions of the flow channels, and Figure 2 shows the geometries of Designs 1–5. Design 1 consisted of serpentine channels with abrupt changes in flow direction, while in Design 2, the direction of the flow was changed gradually. Figure 2A,B presents two versions of the serpentine flow field design. The electrical current distributions of Designs 1 and 2 were compared, and the best one was chosen. The dimensions of the selected design were then modified (Design 3), and once more, the performances of the new flow field geometry were compared with the other designs. See Figure 2C. Design 4 was based on continually re-directing the flow in a crisscross pattern, and it is shown in Figure 2D. The air flow was introduced via three intakes, each one equidistant from one another. Design 4 was developed after a series of design iterations, each one providing a major leap in performance over the previous one. On the other hand, Design 5 used fifteen multiple parallel and straight channels (Figure 2E), and each channel was fed with air at a constant pressure. In this design, the air flow was introduced via an intake located at the center of each group of five channels [25,26].

**Table 4.** Boundary conditions.

| Name | Equations | Number |
|---|:---:|:---:|
| Navier–Stokes Normal Velocity Vector | $u \bullet n = u_0$ | (7) |
| Navier–Stokes Outlet Pressure | $p = p_0$ | (8) |
| Navier–Stokes Velocity Next to the Wall | $u = 0$ | (9) |
| Stefan–Maxwell Convective Flux | $n_i \bullet n = (\rho\omega_i u) \bullet n$ | (10) |
| Stefan–Maxwell Velocity in the Pore | $u = \dfrac{-n_{N2}}{\rho\overline{\omega}_{N2}}$ | (11) |
| Electrochemical Reaction Oxygen | $n_{O_2} \bullet n = M_{0_2}\dfrac{i_c}{4F}$ | (12) |

Note: $n$, mass flux vector mol/s; $u_0$, inlet velocity $= 50$ cc/min or 200 cc/min, $p_0$, outlet pressure $= 1.013 \times 10^5$ Pa.

**Table 5.** Dimensions of the designs used for this study for the simulation.

| Parameter | Design 1 | Design 2 | Design 3 | Design 4 | Design 5 |
|---|---|---|---|---|---|
| Channel width | 1 mm | 1 mm | 1 mm | 1 mm | 1 mm |
| Channel depth | 1 mm | 1 mm | 1 mm | 1 mm | 1 mm |
| Rib width | 1 mm | 1 mm | 1 mm | $1 \times 8 \times 1$ mm | 1 mm |
| Diffusion layer geometric area | 12.32 cm$^2$ | 12.32 cm$^2$ | 31.8 cm$^2$ | 31.8 cm$^2$ | 31.8 cm$^2$ |
| Diffusion layer thickness | 0.2 mm | 0.2 mm | 0.2 mm | 0.2 mm | 0.2 mm |
| Geometric area for fluid flow | 6.16 cm$^2$ | 5.97 cm$^2$ | 15.19 cm$^2$ | 16.27 cm$^2$ | 15.9 cm$^2$ |
| Channel length | ——— | ——— | 152.73 cm | ——— | 12.6 cm |
| Flow/mechanical support relation | 50%/50% | 52%/58% | 53%/47% | 49%/51% | 42%/58% |

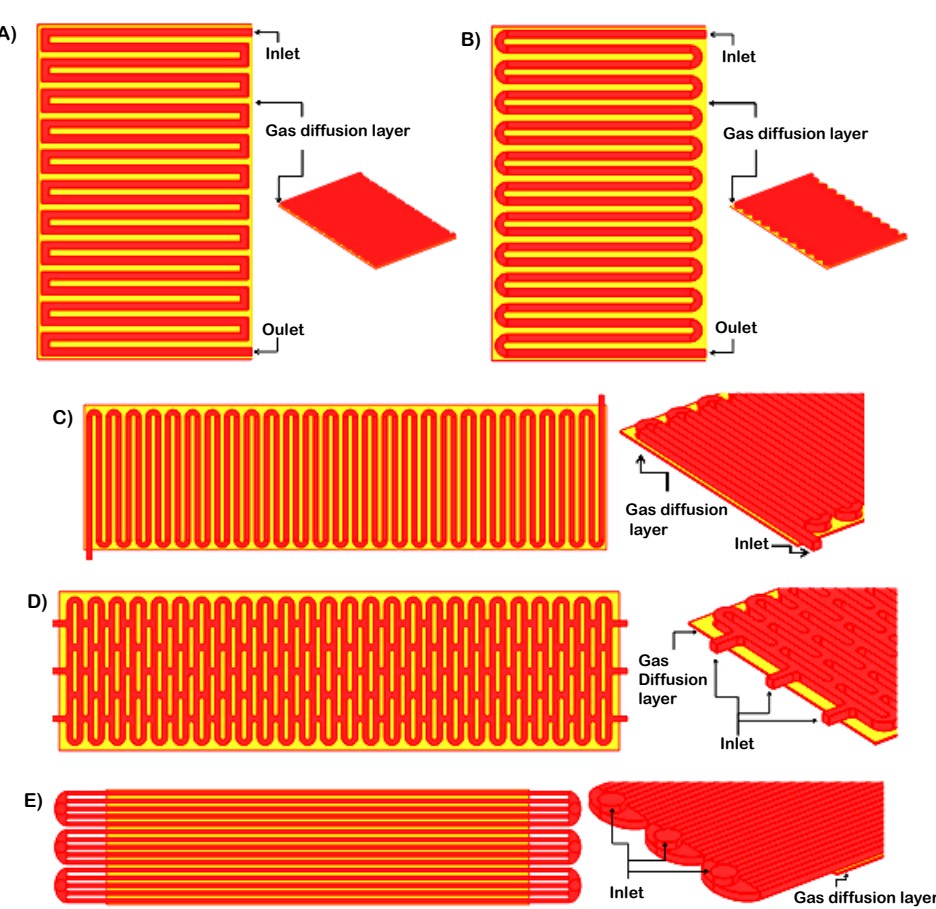

**Figure 2.** (**A**) Design 1 consisted of serpentine channels with abrupt changes in flow direction. (**B**) Design 2 consisted of serpentine channel with a more gradual change in flow direction; (**C**) Design 3 (serpentine) consisted of curvilinear bends; (**D**) Design 4 consisted of a crisscross pattern; and (**E**) Design 5 was made with straight and parallel channels.

## 3. Results

Figure 2 shows the measured gas velocity in Designs 1 and 2. A very slow flow velocity appeared at the square corners when the flow changed direction abruptly (Figure 3A). Therefore, water could accumulate here, which would lead to uneven gas and current distributions. A very slow flow-velocity region was not observed when the flow direction changed gradually in Design 2. Figure 3B–D presents the electrical current distributions in both designs at high cathode overpotential (0.8 V). The porous media mitigated the effect of the slow flow-velocity regions, and the distributions of the current

density were similar in both geometries. From computed polarization curves, it was found that at low and medium power demands, both flow field designs gave basically the same performance. At a higher power demand, the performances differed, but only slightly.

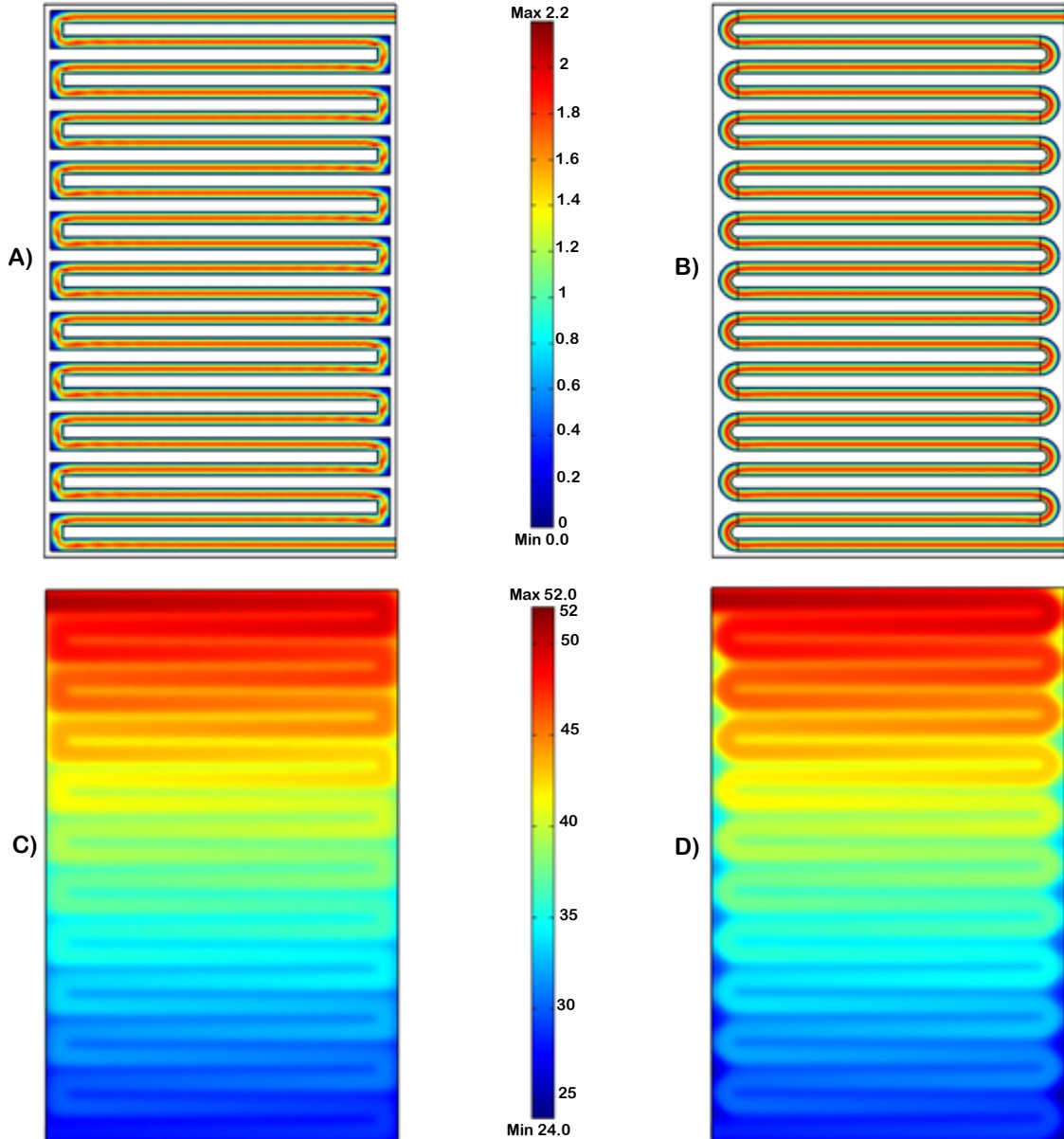

**Figure 3.** Flow velocity within the channels (m/s) in (**A**) Design 1 and (**B**)) Design 2. Air introduced via an intake located at the top right side (inlet). The electrical current density distribution obtained on the bottom of the porous cathode for (**C**) Design 1 and (**D**) Design 2. In this case, the air was introduced via an intake located at the top left side (inlet).

Figure 4 shows the electrical current variation in the y axis, which is perpendicular to the downstream channel direction (Table 2). The wave shape of the current variation was correlated with the lengths of both the channel and the rib [27,28]. This figure shows that the electrical current decreased gradually along the y axis with the maximum value near the inlet. For example, when the distance was 4.4 cm, the total electrical current at 0.8 V was 27.28 mA in Design 1 and 27.69 mA in Design 2. The difference in the total electrical currents between the two designs was 1.48%.

Although this is a negligible difference between designs in one cell, this translates into a large difference in a stack configuration.

A comparison between these designs in terms of pressure drop with increasing Reynolds number [22,29] was made. According to some authors, the serpentine square bends (Design 1) exhibited consistently higher pressure drops compared to the curvilinear bends used in Design 2. A large pressure drop leads to inefficient fuel cell performance. Furthermore, because fuel starvation was more likely to occur in Design 1, Design 2 was selected as the superior standard. Design 2 was therefore modified to create Design 3; see Figure 2C.

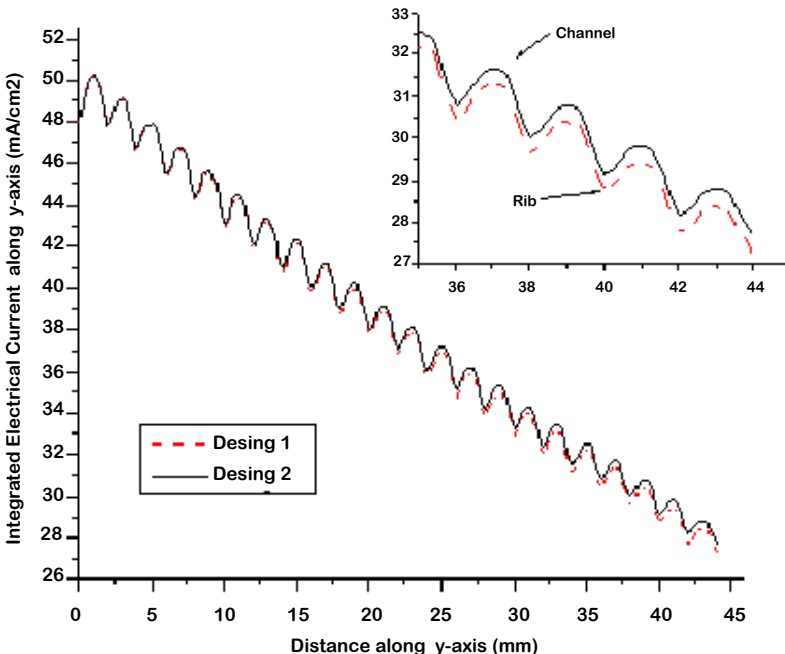

**Figure 4.** The integrated electrical current along the y-axis, where the initial point of the trajectory is located in the middle of the flow field ($x$ = 3.08 cm). Current density obtained in Designs 1 and 2 at a cathode overpotential of 0.8 V.

The oxygen mass fraction distributions for Designs 3–5 are shown in Figure 5. These appeared to be more uniform in Designs 4 or 5 than in Design 3. At a cathode overpotential of 0.66 V, the final oxygen mass fraction in the gas diffusion layer was slightly lower than 0.123 in Design 3. In contrast, in Designs 4 and 5, the final oxygen mass fractions were slightly over 0.137. Again, this small difference observed in a single cell can be significant in a stack configuration [23,24]. Figure 6 shows the current density distributions for Designs 3–5 under a cell voltage of 0.52 V (i.e., a cathode overpotential of 0.66 V). The highest current density was obtained at the inlet where oxygen concentration was the highest. Figure 6 clearly shows that the electrical current density in the outlet was lower in Design 3, and consequently, a less efficient cell performance will be expected. In other words, Designs 4 and 5 produced more uniform electrical current density distributions over the porous cathode than Design 3. Consequently, Designs 4 and 5 could deliver superior performances. At low electrical demands, the performance of all fuel cells was governed by the electrode kinetics (Equation (6)). In this case, the differences among gas-distribution designs were minimal. However, at medium electrical demands (Table 6), the mass transport could not keep up with the moderately fast kinetics, and the performance of Design 3 became mass-transport-limited. At higher electrical demands, the performance of Design 3 deteriorated further (Table 6). For example, at an overpotential of 0.82 V, the total current (1811 mA) was significantly smaller than the total electrical current for Design 5 (2127 mA) [30,31].

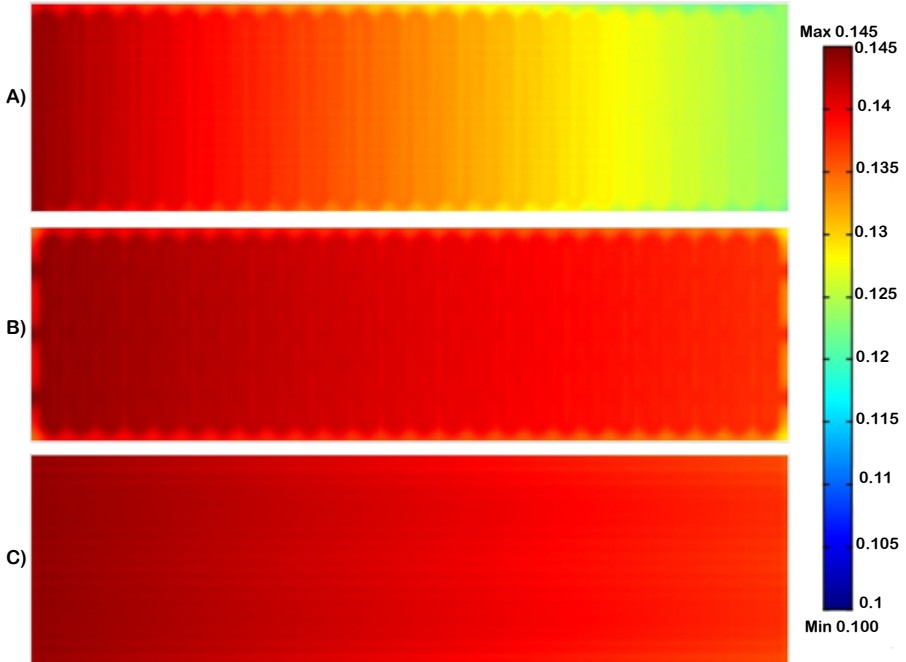

**Figure 5.** Mass oxygen fraction. (**A**) Design 3 (serpentine with curvilinear bends); (**B**) Design 4 (crisscross); and (**C**) Design 5 (straight and parallel channels).

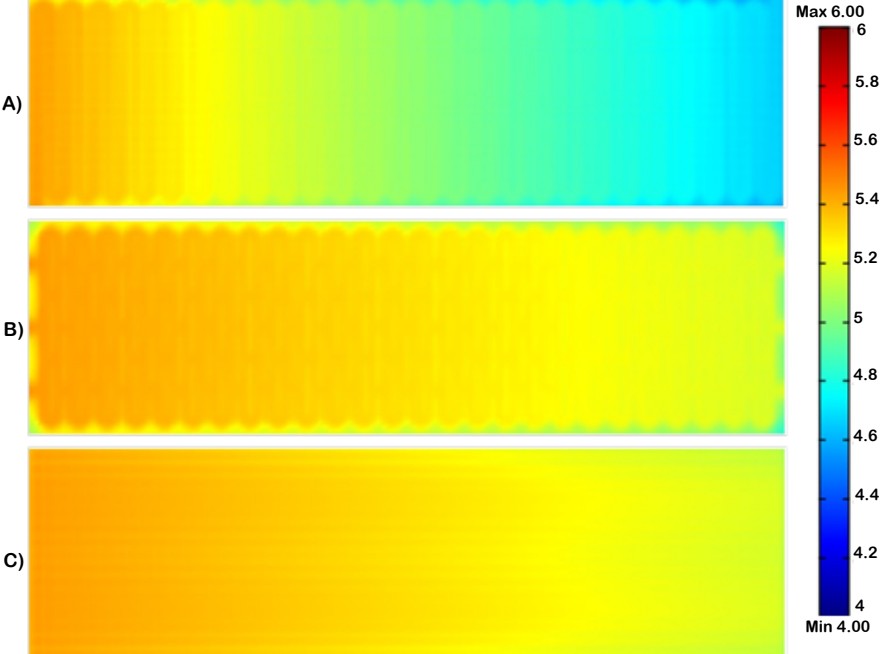

**Figure 6.** Electrical current distributions (mA/cm$^2$) at the cathode (overpotential 0.66 V). (**A**) Design 3 (serpentine with curvilinear bends); (**B**) Design 4 (crisscross); and (**C**) Design 5 (straight and parallel channels).

**Table 6.** Total current for different designs under various operating conditions (mA).

| Design | 50 cc/min 0.66 V | 200 cc/min 0.22 V | 200 cc/min 0.52 V | 200 cc/min 0.82 V |
|--------|------------------|-------------------|-------------------|-------------------|
| 3 | 160.2 | 0.125 | 17.33 | 1811 |
| 4 | 168.4 | 0.125 | 17.35 | 2092 |
| 5 | 168.7 | 0.125 | 17.35 | 2127 |

## 4. Discussion and Conclusions

### 4.1. Discussion

As mentioned in this work, by means of numerical simulations, five designs of bipolar plate geometries of fuel cells were made and compared to different air flow fields adjacent to a PEMFC oxygen cathode. Based on the obtained results, we can mention that in Designs 1 and 2 (Figure 2), the flow rate observed was very slow at the square corners when the flow changed direction abruptly (Figure 2A). In this sense, it is possible to suppose that the water of the system could accumulate in the mentioned corners, and consequently, an uneven distribution of gas and current can be observed. A region of very slow flow velocity was not observed when the direction of flow gradually changed (Figure 2B). Figure 2C,D presents the electric current distributions at a high potential of a high cathode (0.8 V). Figure 3 shows the variation of electric current in the axis, and the electric current gradually decreased along the axis and with the maximum value near the input. Several reports showed the comparisons of designs in terms of pressure drop with the increase in the Reynolds number (Equation (9)). According to Boddu [31] and Rivas [32], for the square serpentine geometries (Design 1) the pressure drops are higher compared to the designs with curves (Design 2). A large pressure drop leads to inefficient fuel cell performance. In addition, because the lack of fuel is more likely to occur in Design 1, Design 2 was selected as the standard to be modified to create Design 3; see Figure 1C. The mass fraction distributions were more uniform in Designs 4 and 5 than in Designs 1, 2, and 3. With a cathode overpotential of 0.66 V, the final mass fraction of oxygen in the gas diffusion layer was slightly below 0.123 in Design 3. In contrast, in Designs 4 and 5, the final oxygen mass fractions were slightly higher than 0.137. In Figure 6, the current density distributions for Designs 3–5 can be observed under a cell voltage of 0.52 V, and the highest current density was obtained at the input where the oxygen concentration was the highest. Figure 6 clearly shows that the electrical current density at the output was lower in Design 3, and consequently, less efficient cell performance was expected. Designs 4 and 5 produced more uniform distributions of electric current density over the porous cathode than Design 3. Therefore, Designs 4 and 5 can offer higher yields. At low electrical demands, the performance of all fuel cells was governed by the kinetics of the electrode (Equation (6)). In this case, the differences between the gas distribution designs were minimal. The performance of Design 3 became limited in mass transport. With higher electrical demands, the performance of Design 3 deteriorated further (Table 5). With an overpotential of 0.82 V, the total current (1811 mA) was significantly less than the total electric current for Design 5 (2127 mA). The distributions of the oxygen mass fraction for Designs 3–5 are shown in Figure 5. These appeared to be more uniform in Design 4 or 5 than in Designs 1, 2, and 3. Barreras [9] and Rivas [33] showed that the profiles of current density exhibited better distributions with the re-direct pattern design, to produce a more efficient fuel cell. Our results were in agreement with those presented by this author. On the other hand, Rivas [33] observed poor current distributions with parallel flow fields. Figure 6 shows the current density distributions for Designs 3–5, for which the cell voltage fell below 0.52 V (that is, 0.66 V of cathode over potential). The highest current density was obtained at the entrance where the oxygen concentration was the highest. Figure 6 clearly shows that the electrical current density at the output was lower in Design 3, and consequently, less efficient cell performance was expected. With an overpotential of 0.82 V, the total current was 1811 mA. It was significantly lower than the total electric current for Design 5 (2127 mA).

Drendel [34] proposed that the external power generation and energy efficiency were considered as indices for PEM performance. The main results established in this work: five designs with different channel shapes and gas flow patterns, were simulated numerically using computational fluid dynamics (CFD). Using this approach, it was possible to conceive of, detail, and analyze each design. The electrical current distributions of the two different serpentine channels with square or rounded flow bends were tested. We can mention some studies that we relied on to arrive at our results. P.Havaej [35] utilized a two-phase, multicomponent, transient, and three-dimensional model for simulating the performance of the PEMFC. In the first step, the best longitudinal catalyst loading distribution was found. In the second step, several lateral distributions were superimposed on the noted longitudinal catalyst loading distribution, and the performance of the PEMFC was evaluated for each distribution. Numerical results showed 3.1% enhancement for the longitudinal catalyst loading distributions. Um [36], through simulation, was able to detail the cell current density response to a step-change in cell voltage.

By studying bipolar plates with different flow channel configurations, through computational modeling of fluid dynamics, a comparison was able to made of the pressure drop characteristics for different flow channel designs. The results showed that with a greater number of parallel channels and smaller sizes, a more effective contact surface area can be achieved along with a decrease in pressure drop. The correlations of these coefficients of the input region will be useful for the fuel cell simulation model. Xun Zhu [37] analyzed different geometry configurations of collective flow plates using CDF and found for microchannels with different cross-section geometries, the detachment time, detachment diameter, and the removal time of water droplets increased in this sequence: triangle < trapezoid < rectangle with a curved bottom wall < rectangle < upside-down trapezoid. The detachment time for semicircle channel was longer than the rectangle, while its detachment diameter was smaller and the removal time shorter than the rectangle. Yan [38], by simulating the flow channels, better uniformity in the current density distribution along the width of the cell could be attained. In other cases, the researchers showed real applications mainly for the automotive sector in hybrid vehicles or applications in the electronics industry such as telephony, computing, and instruments for research and development [33]. Zamora, using ANSYS $^{®}$ v14, simulated the behavior of the relevant variables in a fuel cell (pressure, volume, mass flow) in eight different geometry configurations in bipolar plates. More uniform results were obtained in a geometry configuration with grooves [39], and the simulation reduced the time for laboratory-scale experimentation. In our own case, we carried out an application at the laboratory level applying polymeric materials for the manufacture of bipolar plates and the construction of a stack, achieving results similar to those obtained in commercial fuel cells, our contribution resulted in a decrease in the number of components (screws), since the manufactured cells were sealed with the same material as the bipolar plates. In most of the cases mentioned, the study was limited to the simulation of the plates and their different geometries [39]. Peng [40], through the design of experiments (DOE) method and an optimization method known as adoptive simulation (ASA), proposed an optimization model of the flow channel section design for a hydroformed metal bipolar plate. The optimization results showed that the optimal dimension values for the depth of the channel, the width of the channel, the width of the rib, and the transition radius were 0.5, 1.0, 1.6, and 0.5 mm, respectively, with the greatest reaction efficiency (79%) and acceptable formability (1.0). Kong developed a model capturing the key geometric parameters and their interrelationship, which were required to derive explicit expressions of the key electrode parameters in fuel cells [41].

### 4.2. Conclusions

Five designs with different channel shapes and gas flow patterns were simulated numerically using computational fluid dynamics (CFD). Using this approach, it was possible to conceive of, detail, and analyze each design. The electrical current distributions of the two different serpentine channels with square or rounded flow bends were tested. At higher loads, the serpentine channels with abrupt changes in flow direction (Design 1) exhibited slightly lower electrical current distributions compared

to curvilinear bends (Design 2). This minute difference within single cells across designs can have a large effect in a stack configuration. It is recommended that the serpentine channel configuration should consist of curvilinear bends. Because of its better performance, Design 2 was selected for the second part of this investigation. The dimensions of the serpentine channel design with curvilinear bends were then modified (Design 3). Afterwards, Design 3 was compared to crisscross (Design 4) and straight parallel channels (Design 5). The results showed that Designs 4 and 5 produced more uniform electrical current distributions than Design 3. This can be explained because the intakes were efficiently located. Therefore, it was concluded that the effective placement of the intakes was as important as the shape of the channel configuration. The parallel channel flow field (Design 5) was the best alternative for current collectors due to its superior performance.

Some of the most important components of a fuel cell are the bipolar plates and the geometries of the channels for the distribution of the gases, on which the efficiency of the cell will depend. Fuel cells are an excellent alternative for the use of renewable energy. Mexico is a developing country, which has the highest demand for electricity in the world with values of 5.8%–6% per year. Increasingly stringent environmental policies across the planet demand prompt solutions to pollution problems and demand lower and lower emission levels. The instability of oil prices is forcing countries like Mexico to stimulate an economy less dependent on this energy, looking for solutions such as the one shown in this paper. Fuel cells are an alternative, but developing this technology is a very expensive process in Mexico and in many countries around the world that are opting for the use of renewable energy. Simulation represents an alternative to continue with the development and applications. Some of its main advantages we can mention and that have motivated us to develop the research work are: (a) it provides many types of possible alternatives to explore; (b) the simulation provides a simpler method of a solution when mathematical procedures are complex and difficult; (c) it provides total control over time, because a phenomenon can be accelerated; (d) in some cases, simulation is the only means to achieve a solution; (e) it is generally less expensive to improve the system via simulation than to do it in the real system; (f) it helps the innovation process since it allows the experimenter to observe and play with the system; (g) it is much simpler to visualize and understand simulation methods than purely analytical methods; it gives a deep understanding of the system; (h) it gives solutions to problems "without" an analytical solution; (i) it allows analyzing the effect on the overall performance of a system; and (j) it allows experimentation in conditions that could be dangerous or that have a high economic cost in the real system. It is important to mention that there are not many reports about similar cases in which the use of the numerical simulations to develop PEM fuel cells has been employed. In this sense, the innovation degree of our work suggest a transcendental relevance in the PEM fuel cell design field. The objective was met.

## 5. Concluding Remarks

Based on the results achieved, it is intended to manufacture prototypes of bipolar plate designs and perform laboratory tests.

**Author Contributions:** Conceptualization, R.G.-G.; data curation, R.G.-G. and E.R.S.; formal analysis, M.A.Z.-A., P.E.O.P., G.O.-G., and J.M.O.-R.; investigation, M.A.Z.-A., P.E.O.P., and E.R.S.; methodology, P.E.O.P. and J.M.O.-R.; project administration, G.O.-G.; software, G.O.-G. and J.M.O.-R.; visualization, M.A.Z.-A.; writing, original draft, R.G.-G.; writing, review and editing, E.R.S.; investigation and writing, review and editing, Á.D.J.R.B.

**Funding:** This research was partial funded by CONACYT FUNDER Grant Number 1028088.

**Acknowledgments:** The authors appreciate the CONACYT Scholarship 1028088 of Pablo Esau Hidalgo-Pimentel to develop this project. The authors would like to thank the *Centro de Investigación y Desarrollo Tecnológico en Electroquímica, S.C.* (CIDETEQ) for their contribution of resources and materials to this research. The authors would like to express their gratitude to Aaron Rodriguez Morales for his advice in the English language.

**Conflicts of Interest:** The authors declare that they have no conflict of interest.

## Abbreviations

The following notations are used in this manuscript:

**Governing Equations**

| | |
|---|---|
| $\rho$ | density kg/m$^3$ |
| t | time s |
| $\mu$ | dynamic viscosity of oxygen Pa |
| $u$ | velocity m/s |
| p | pressure Pa |
| $\omega_i$ | weight fraction of the i$^{th}$ species; with inlet mass fraction O$_2$, mass fraction of H$_2$O |

**Stefan–Maxwell Equations**

| | |
|---|---|
| $D_{ij}$ | binary diffusion coefficient for species i and j |
| $D_{eff,ij}$ | effective binary diffusion coefficient K |
| $v_i$ | molar diffusion volume of the j$^{th}$ species cm$^3$/mol |
| M | total molar mass of the mixture g/mol |
| Maxwell diffusion constant | $3.16 \times 10^8$ Pa$\delta$m$^2$/s |
| $M_j$ | molecular weight of species j g/mol |
| $MO_2$ | molecular weight of O$_2$ g/mol |
| $MH_2O$ | molecular weight of H$_2$O g/mol |
| $MN_2$ | molecular weight of N$_2$ g/mol |
| T | temperature |
| $vO_2$ | molar diffusion volume of O$_2$ cm$^3$/mol |
| $vH_2O$ | molar diffusion volume of H$_2$O cm$^3$/mol |
| $vN_2$ | molar diffusion volume of N$_2$ cm$^3$/mol |
| $\varepsilon$ | porosity |
| $i_c$ | cathode current |
| $S_a$ | specific surface area m$^2$/m$^3$ |
| $\delta$ | active layer thickness |
| $i_0$ | exchange current density mA/cm$^2$ |
| F | Faraday's constant = 96,485 C/mol |
| R | gas constant = 8.314 J/mol |
| $\eta$ | overpotential |
| $t_{H_2O}$ | electro-osmotic drag |
| gdl | gas diffusion layer thickness |

**Navier–Stokes Equations**

| | |
|---|---|
| n | mass flux vector mol/s |
| $u_0$ | inlet velocity |
| $p_0$ | outlet pressure Pa |

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
