# Peer review of "Flow Analysis Based on Cathodic Current Using Different Designs of Channel Distribution In PEM Fuel Cells"

_applsci, doi:10.3390/app9173615_

Round 1

Reviewer 1 Report

This paper addresses a development of flow channel design for PEMFCs. Overall, this paper deserves publication. The reviewer recommends that the authors add a discussion for applicability of the geometries of the flow fields analysed in the paper to actual and practical PEMFC stacks.

Title of the paper should be revised as "FLOW ANALYSIS BASED ON CATHODIC CURRENT USING DIFFERENTS DESINGS OF CHANNEL DISTRIBUTION IN PEM FUEL CELLS PEM. Some typographical errors found in the paper should be corrected.   

Author Response

This paper addresses a development of flow channel design for PEMFCs. Overall, this paper deserves publication. 

Point A. Title of the paper should be revised as "FLOW ANALYSIS BASED ON CATHODIC CURRENT USING DIFFERENTS DESINGS OF CHANNEL DISTRIBUTION IN PEM FUEL CELLS PEM. Some typographical errors found in the paper should be corrected.  

A: Thank you for your comments. In the new version of the paper we are specifying the changes in the Title such as "FLOW ANALYSIS BASED ON CATHODIC CURRENT USING DIFFERENTS DESINGS OF CHANNEL DISTRIBUTION IN PEM FUEL CELLS.

Point B. The reviewer recommends that the authors add a discussion for applicability of the geometries of the flow fields analysed in the paper to actual and practical PEMFC stacks.

B: Thank you for your comments. In the new version of the paper, we rearrange the Discussion part as the reviewer suggested add a discussion for applicability of the geometries of the flow fields analyzed in the paper to actual and practical PEMFC stacks.

In the new version of the paper, we are add the section

Futures Works:    Based on the results obtained, we intend to manufacture the prototypes of bipolar plate designs and perform tests at the laboratory level.

Reviewer 2 Report

1. The title part and table 4, please clearly explain the meaning of "desing" . 2. It is better to include a detailed comparison of similar studies with the literature. 3. It is better to provide up-to-date literature review. 4. It is better to state specific objectives. 5. It is better to include some physical experiments to confirm the simulation result. 6. It is better to explain why the currents of three designs are almost the same when the overpotential is not 0.82 eV.

Author Response

The title part and table 4, please clearly explain the meaning of "desing"

A: Thank you for your comments. In the new version, In the Introduction section the definition of the term “design” was added, which refers to the shape of the geometry of the flow channels of the bipolar plates analyzed in this work and Table 1 was added with the characteristics of each design of geometries of the bipolar plates analyzed in the present work. See lines 42 to 46. We include the table to explain what each design consists of.

2. It is better to include a detailed comparison of similar studies with the literature.  

B: Thank you for your comments. In the new version of the paper, we rearrange the discussion part as the reviewer suggested add a discussion for applicability of the geometries of the flow fields analyzed in the paper to actual and practical PEMFC stacks.

 3.It is better to provide up-to-date literature review

C: Thank you for your comments. In the new version of the paper, we rearrange the Discussion part as the reviewer suggested add a discussion for applicability of the geometries of the flow fields analyzed in the paper to actual and practical PEMFC stacks.

4.It is better to state specific objectives

D: Thank you for your comments. In the new version of the paper, we are adding the section Objective:

The main focus of this study was to measure the electrical current distribution in these designs to find an efficient and a commercially viable PEMFC, and to determine the best option for a possible development of a fuel cell at the laboratory level from the results achieved.

 5. It is better to include some physical experiments to confirm the simulation result.

D: Thank you for your comments.  We are in a fundraising process to be able to make the prototypes and continue our projects. Currently, the Consejo Nacional de Ciencia y Tecnología (CONACYT) has modified policies to grant financial support and resources for research. Almost all calls for resources have been stopped.  The guideline of the New Government of México is focused on saving resources in all areas and sectors of Public Administration. Our Institution Universidad del Valle de México within its policies for research and development will support us with economic resources to the extent that we can count on high impact scientific populations.

In the new version of the paper, we are adding the section Futures Works.

Futures Works:   Based on the results achieved, it is intended to manufacture prototypes of bipolar plate designs and perform tests at the laboratory level.

6. It is better to explain why the currents of three designs are almost the same when the overpotential is not 0.82 eV. 

 E: Thank you for your comments In the new version of the paper. Different parameters were applied in terms of volume, oxygen, current, etc., in each of the designs. Designs 1, 2 and 3 are similar. Design 3, was made from designs 1 and 2.   The study is a simulation that allows modifying and analyzing the designs using different values and parameters. In the results in lines  148 to 165.

7.The manuscript should be checked by a native speaker.

F: Thank you for your comments. A native speaker has reviewed the new version of our paper.

Round 2

Reviewer 2 Report

This revision is ready to be published.

Author Response

Please check section numbering

Thank you very much for the comments. In the new version of the paper already adjusted the numbering.

Please avoid using collective references (e.g. [2-10]). Papers should be cited one by one showing what is new in the present submission with respect to the findings available in literature.

Thank you very much for the observation, in the new version of the paper we have made the relevant adjustments with the references.

Authors must significantly improve and update literature review paying special attention to the papers published in this Journal regarding this topic. It seems that only a few paper by this Journal are cited by the authors. If this is the case, the present submission would be better suitable for an other type of Journal.

Thanks for the observation. We have reviewed and updated the literature and updated the references in the new version of the paper.

Authors must improve the paragraph proving the novelty and the scientific relevance of their work with respect to the findings available in literature. They must clearly respond to those questions:  what is new? What about the scientific relevance of this work?

Thanks a lot for the suggestion. In the new version of the article, we have made the adjustments.

In this type of scientific work, each finding is an innovation to the theme of configurations in bipolar plate geometries, our contribution is a new configuration of serpentine geometry. The scientific relevance lies in the new modifications applied to the serpentine geometries of fuel cell collector plates.

In the new version of the article  we have included the following paragraph in the lines 27 to 40:

Actually, a large number of Companies, Education Institutions, and Research Centers are currently developing research programs in fuel cells. However, there are still a significant amount of technical challenges that must be addressed, such as the choice and handling of fuels, the lack of infrastructure for hydrogen storage and the high cost of this, etc. The technology of the Fuel cells is sufficiently developed for its commercialization, except for the cost, it is still very high. Therefore, the most important technological and scientific activity in the development of fuel cells is aimed at reducing costs and improving their performance. The development and application of new configurations of collector plate geometries is a fundamental aspect to improve the efficiency of fuel cells. In the literature, many researchers continually report new findings and proposals for improvement or new geometries, which allow different aspects of technology and fuel cell efficiency to be improved, but they are still not enough to reduce the costs involved in the development of this technology.  In the following years, it is expected to reduce the cost of manufacturing fuel cells so that they are economically viable and mass production can occur, at least. In the short term, it is expected that the fuel cell application will be present in the power generation and remote power distribution systems.

Section Objective should be moved within introduction and/or expanded showing the novelty

Thanks a lot for the suggestion. The objective has been included in the introduction. See Lines 72 to 74.

What about model validation?

We include the section 4 Concluding Remarks. The next stage of research will be the construction of a prototype for validation at the laboratory level.

What about mesh sensitivity analysis?

A simulation is a tool that allows you to vary the meshes nodes in experimentation of the type we are presenting. It starts with three-node meshes and can be expanded almost infinitely.  The mesh is an input parameter in the process of designing bipolar plate geometries in Fuel Cells. Its mathematical validation depends on the support of simulation tools such as ANSYS or COMSOL, to perform the finite element analysis. It is simply the mathematical solution to the equations that support the phenomena that can occur in the behavior of a fuel cell, for example, the Navier Stoke state equations, solve the problem of pressure drop, for our case we use the equations by Stefan Maxwell to analyze the behavior of electric current. Without the aforementioned tools of ANSYS or COMSOL, it would be almost impossible to solve the statements made regarding the solution of an almost infinite number of differential equations, there are no cases supported in the literature in almost 20 years, where a researcher has been given the task of solving a system of equations for fuel cells freehand. Through the simulation, input values ​​are defined for the relevant variables of the system to be analyzed, such as Pressure (p), Mass Flow, or voltage (V). With the figures No 2 and 5 reported in the  work, validation was carried out, and the most efficient geometry was determined. The flow field in the design of bipolar plates is another important feature for the proper functioning of Fuel Cells and there are a wide variety of types of flow fields for bipolar plates.

As reported in the literature about flow fields for bipolar plates, the performance of a fuel cell depends on a series of parameters such as plate geometry, flow field, fluid pressure, etc. The analysis of the literature and the operability of the COMSOL software allowed defining some criteria to select the profiles of the channels on the plates. Likewise, the selected flow field will be incorporated into a PEM type fuel cell computational model to analyze its efficiency in energy conversion. Criteria: 1 Select an easy-to-manufacture geometry with the equipment currently available. For example, serpentine multiple serpentine flow fields, parallel channels, and parallel models are relatively easy to manufacture. 2 Geometry of easy implementation in a fuel cell model for analysis with the finite element method and avoid those that due to their complex design represent a great challenge and are difficult to manipulate due to the computational capacity available in these moments, despite the attractiveness of their behavior. 3 Efficiency in the distribution of fuel throughout the entire area of ​​the flow field, minimizing stagnation areas. 4 Pressure drop less than the coil type model.

The geometries of the flow fields that were simulated in this work have the following modifications for those reported in the literature: 1. Sizes and shapes of obstacles to the passage of fuel 2. Position of obstacles to the passage of fuel to cover a larger area. 3. Positions of the entry and exit of gases 4. Curvature in the coils 5. Distribution of the channels, in the parallel channel model and the coil. The meshing process was carried out with the COMSOL Meshing module, with the CFD mesh option for Fluent. The size of the element selected for these flow fields was 0.1 mm, and the types of elements selected for the model with straight edges were hexahedra (8 nodes) and tetrahedra (4 nodes), while hexahedrons were selected in the coil with rounded curves (8 nodes). We consider it not necessary to include this paragraph because the students of the subject know it.

A nomenclature including all the parameters and the related units must be provided (is it section 6?? The title should be changed.

Thank you very much for the comments. The section 6 change in the new version of the paper already adjusted the title of section No 5 per Glossary.  Section 6 change to section 5.

Authors must carefully check the format and the text: some typos are included in the text (please check also the title).

Thank you very much for the comments. In the new version of the paper, we have already made the suggested changes in the title

Flow Analysis based on Cathodic Current using Different Designs of Channel Distribution in PEM Fuel Cells

This manuscript is a resubmission of an earlier submission. The following is a list of the peer review reports and author responses from that submission.